# Reduction of bioburden on large area surfaces through use of a supplemental residual antimicrobial paint

Jennifer Hiras[1], Kelly R. Bright[2], Jackie L. Kurzejewski[1], Alexander E. McInroy[1], Anthony G. Frutos[1], Mark R. Langille[1], Jon Q. Lehman[2], Charles P. Gerba[2], Joydeep Lahiri[1] *

**1** Corning Incorporated, Corning, NY, United States of America, **2** Department of Environmental Science, The University of Arizona, Tucson, Arizona, United States of America

* lahirij@corning.com

**Data Availability Statement:** All relevant data are within the manuscript and its Supporting Information files.

**Funding:** Corning Incorporated provided support in the form of salaries for authors [JH, JK, AM, ML,

## Abstract

Paint is a versatile material that can be used to coat surfaces for which routine disinfection practices may be lacking. EPA-registered copper-containing supplemental residual antimicrobial paints could be used to reduce the bioburden on often-neglected surfaces. An interventional study was conducted by painting the walls of a preschool restroom and metal locker surfaces in two hospital locker rooms with a copper-containing antimicrobial paint to evaluate the potential for bioburden reduction compared to a non-copper-containing control paint. The antimicrobial paint reduced the bioburden on the preschool restroom walls by 57% and on lockers in one locker room by 63% compared to the control paint; no significant difference was observed between the two paint types in the second locker room. The upper quartile bacterial counts, which drive the overall risk by increasing exposure to pathogens, also exhibited 63% and 47% reductions for the antimicrobial paint compared to the control paint in the preschool restroom and the first locker room, respectively. Because detectable levels of bioburden are found on large-area surfaces such as walls and lockers, surfaces painted with copper-containing paints may make large-area surfaces that are prone to contamination safer in a way that is practical and economical.

## Introduction

Bioburden on surfaces can contribute to the transmission of harmful pathogens. Contamination of inanimate objects (fomites) has been studied extensively, particularly for high-touch surfaces such as mobile phones [1], keyboards [2], and touch screens [3]; however, other large-area surfaces should be considered as part of comprehensive disinfection plans. For instance, floors have been identified as a potential source of pathogen transmission, particularly because floors are typically cleaned but not disinfected [4,5]. Implementing daily disinfection practices is often impractical; rather, imparting these surfaces with residual antimicrobial activity offers a pragmatic alternative. We present here the first interventional study for large-area surfaces

AF, JL], but did not have any additional role in the study design, data collection and analysis, decision to publish, or preparation of the manuscript. The specific roles of these authors are articulated in the "author contributions" section. The University of Arizona received fees for services rendered for the processing and assay of the collected samples for the detection and quantification of HPC bacteria, coliforms, and Escherichia coli.

**Competing interests:** The authors [JH, JK, AM, ML, AF, JL] are employees of Corning Incorporated. This does not alter our adherence to PLOS ONE policies on sharing data and materials. Corning Incorporated has commercialized the copper-glass ceramic additive (Corning® Guardiant®) used in the antimicrobial paint reported.

painted with a copper-containing antimicrobial paint that provides supplemental residual antimicrobial activity.

Paint is a versatile and long-lasting coating that can be applied to arbitrary surfaces. With the right primer, an antimicrobial paint can be applied to drywall, wood, metal, or plastic, thereby offering a flexible and economical approach to lowering bioburden on surfaces that may be overlooked by routine cleaning and disinfection practices. Copper metal is a well-studied antimicrobial material capable of killing bacteria and reducing hospital-acquired infections [6–8]. The U.S. Environmental Protection Agency (EPA) recently approved copper-containing antimicrobial paints that are continuously active and can provide efficacy between regular cleaning and disinfection practices, including efficacy against SARS-CoV-2, the coronavirus that causes COVID-19 [9]. The antimicrobial efficacy of these paints is imparted through use of a copper-containing glass-ceramic additive containing cuprous ions ($Cu^{+1}$) in a water-labile phase that enables the potency of metallic copper without its look and feel [10].

In the current study, two applications of the antimicrobial paint were considered: (1) restroom walls in a preschool and (2) lockers in a hospital. Though walls demonstrate lower overall levels of contamination than high-touch surfaces, we have found higher contamination levels at lower heights (See section Preliminary study on hospital wall contamination). This finding prompted our investigation of restroom walls at a preschool, given the propensity of children to touch walls. Lockers were chosen as an example of large-area surfaces that are touched at moderate levels but are typically neglected for routine disinfection. We investigated the impact of coating lockers with an antimicrobial paint at a local hospital. Previous studies have shown that reduction of heterotrophic plate count (HPC) bacteria on surfaces containing residual antimicrobial activity by 64% to 85% can result in a significant decrease in healthcare associated infections [6,7,11].

## Materials and methods

### Organic contamination screening

UltraSnap adenosine triphosphate (ATP) swabs were used along with a SystemSURE Plus luminometer (Hygiena, Scarborough, MA) to conduct an initial organic material screen on 4"x4" (103.2 cm$^2$) areas. Follow-up assessments of pre-intervention bioburden levels were conducted by swabbing target surfaces and culturing for bacteria (see section Microbial methods).

### Preliminary study on hospital wall contamination

In prior work conducted at a different hospital, evaluations of walls found higher levels of bioburden at lower heights near the floor. Swabs from walls (waiting rooms, hallways, patient rooms) were collected at different heights over 22 weeks and quantified for bioburden (**Fig 1**). Samples collected at heights >2 feet (2.5–6 feet, average = 3.5 feet, n = 108) had a median bacterial count of 172 Colony Forming Units (CFU)/100 cm$^2$, while samples collected at 0.33 feet (n = 64) had a median count of 378 CFU/100 cm$^2$, a greater than two-fold increase in median bacterial bioburden.

### Intervention

Interventional studies were conducted at a preschool facility and a hospital in western New York state from February-October 2023. The study involved re-painting target surfaces: preschool restroom walls and hospital locker surfaces. These represent typical environments for which supplemental residual antimicrobial products are intended. Paints were purchased from a hardware store. Half of the surfaces were painted with an EPA-registered supplemental

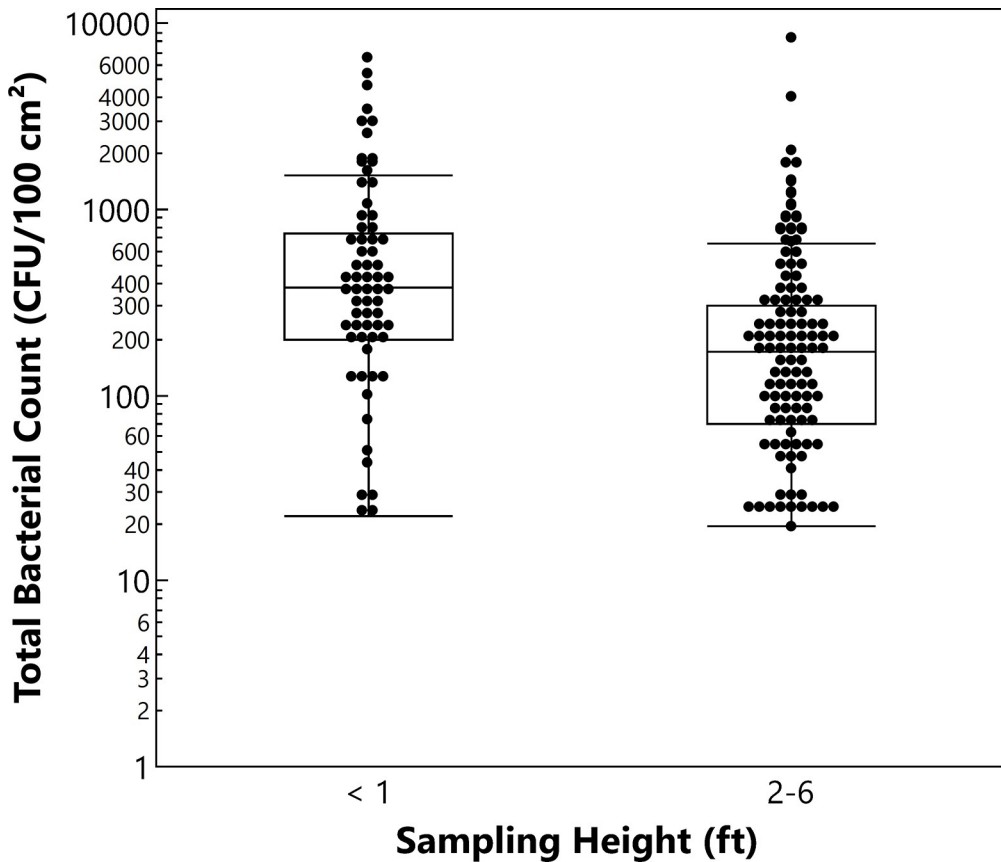

**Fig 1. Total bacterial counts measured from hospital walls at different heights above the floor.** The box and whisker plot displays the count distribution: Median (line within box), quartiles (box top and bottom lines = upper and lower limits of 1st and 4th quartiles, respectively), and outliers (beyond whiskers).

residual antimicrobial paint (Copper Armor™, EPA Registration No. 56601–4; PPG Industries, Inc., Pittsburgh, PA) and half with a traditional, non-antimicrobial paint (Glidden® Diamond®; PPG Industries, Inc.). Both paints were tinted at the point-of-sale to match the existing color of the original painted surface. Paints were applied by professional painters using traditional equipment (e.g., rollers, brushes). To ensure good adhesion to the lockers, a primer (STIX® Waterborne Bonding Primer; Benjamin Moore & Co., Montvale, NJ) was applied prior to the paint application.

## Study design

At the preschool, two adjacent restroom walls and the interior restroom door were painted with the antimicrobial paint; the two opposite walls were painted with the control paint. Samples were collected for bacterial culturing every 3–4 weeks for 5 months on weekdays before the start of the school day. Surfaces were swabbed at a height of 2–3 feet above the ground. While many restroom objects (e.g., sink, toilet, mirror) were routinely cleaned, there was no established cleaning schedule for the walls. The preschool was occupied by approximately 27–30 children/day (aged 3–10 years-old), plus adult staff, and was open five days/week. The restroom used for the intervention was the primary restroom used by the children.

At the hospital, the men's and women's locker rooms utilized by hospital staff were painted. These locker rooms contained 52 and 65 lockers, respectively, with two bays in each. One

locker bay was painted with the antimicrobial paint (27 lockers in the men's room, 41 lockers in the women's room); the other locker bay was painted with the control paint (25 lockers in the men's room, 24 lockers in the women's room). Samples were collected every month for six months for bioburden, coliform, and *E. coli* counts. The lockers were sampled in the mornings (8:00–10:00 a.m.), when use was at lower-than-average capacity to minimize disruption to hospital staff. The locker surfaces were sampled at the lower vents, which consisted of six 6"x1" slats located 6"-12" above the floor. This location was chosen based on previous results (Fig 1). The locker room usage was not tracked during the study, but an estimated 70–80 men and 90–100 women used the respective rooms. As part of the sampling protocol, the hospital lockers were disinfected with isopropyl alcohol wipes immediately after sample collection. To remove dust/debris buildup, all lockers were disinfected one week prior to sample collection in month four. Given the negligible risk with applying commercially available architectural paints and the lack of data collection from human subjects, the management at the daycare and the management at the hospital determined that Institutional Review Board (IRB) approval was not required for this study. Those using the facilities were unaware of the type of paint applied to each surface; normal use and cleaning schedules were maintained. Cleaning and disinfection schedules were not known for the study areas.

Numerous controls were included in the interventional study. In addition to the non-antimicrobial control paint, copper metal and stainless-steel coupons were used as environmental controls. Sampling also occurred on low-touch painted surfaces (locker surfaces 4–6 feet above the floor) and high-touch unpainted surfaces (plastic locker handles) as additional controls to ensure integrity of the collected data. Experimental controls were also included during sampling, which consisted of a spike of *E. coli* (ATCC#11229) containing $1 \times 10^7$ cells in 20 µl onto antimicrobial and control painted surfaces. Spiked surfaces were air dried for 30 minutes before sample collection.

## Microbial methods

Sponge-Sticks (3M, Minneapolis, MN) containing 10 ml of Dey-Engley (D/E) neutralizing broth were used to sample areas of 6"x6" (232.3 cm$^2$) or 8"x12" (619.4 cm$^2$). Both sides of the Sponge-Stick were swabbed back and forth three times per side for each sample. The samples were sent overnight on ice to a university laboratory for culturing. The liquid from each Sponge-Stick was harvested and transferred to sterile 15-ml polypropylene tubes and the sample volume recorded (~5–7 ml).

The samples were vortexed for 10 seconds at high speed. A 4-ml aliquot was removed and added to 96 ml of sterile phosphate buffered saline (PBS; pH 7.4) (Sigma-Aldrich, Inc., St Louis, MO) and assayed via the Colilert® system in Quanti-Tray®/2000 trays (IDEXX laboratories, Inc., Westbrook, ME). The sealed trays were incubated for 24 hours at 37˚C. Sample wells were compared to a positive control inoculated with *E. coli* (ATCC#25922). Yellow wells were considered positive for total coliforms; yellow wells that fluoresced under UV light (365 nm) were considered positive for *E. coli*. The numbers of positive wells were compared to a Colilert® Quanti-Tray®/2000 Most Probable Number (MPN) table to determine the MPN/100 ml (= MPN/4 ml of sample). The MPN/4 ml was adjusted to the MPN recovered per sample (based on the volume recovered).

From the remaining sample, serial 10-fold dilutions were performed using sterile PBS and 100-µl volumes of each dilution were inoculated onto duplicate R2A agar (Becton Dickinson, Franklin Lakes, NJ) plates using the spread plate method. The plates were incubated at 30˚C for five days. Resulting colonies were counted to determine the colony forming units (CFU) of HPC bacteria/ml of sample. This count was adjusted to determine the CFU/entire sample

(based on the volume recovered). All bacterial numbers (MPN or CFU) were then adjusted based on the surface area sampled to determine the MPN or CFU recovered per 100 cm$^2$.

## Data analyses

Not including controls, a total of 89 samples were collected from the preschool restroom and 186 samples from the hospital locker rooms from painted surfaces. An additional 20 and 54 controls (i.e., blanks, spiked controls, unpainted high-touch controls, painted low-touch controls) were collected from the preschool restroom and hospital lockers, respectively. All bioburden samples were analyzed by a non-parametric Wilcoxon test to better handle the right-skewed data. Differences between bacterial levels were considered statistically significant if $P$ was ≤0.05. No measured samples were omitted from the analysis, except when specifically investigating the upper quartiles of the data. All analyses were performed using JMP (v16.2) statistical graphing software (SAS Institute, Inc., Cary, NC).

## Results

### Preschool restroom studies

At all intervention locations, surfaces were first screened for organic contamination using ATP swabs because of their speed and convenience. At the preschool, the level of organic material detected on the restroom walls (average = 73 relative light units (RLU), n = 27) was similar to other surfaces such as windowsills, cabinets, drawers, and drawer handles (average = 74 RLU, n = 62). Thus, the interventional study evaluated contamination levels after the coating of paint was applied onto restroom walls.

The results of bacterial contamination over the four-month study at the preschool restroom are shown in **Fig 2**. The median HPC bacterial count for the walls with the antimicrobial paint was 204 CFU/100 cm$^2$, while the walls with the control paint had a median count of 494 CFU/100 cm$^2$. The difference was statistically significant ($P$ = 0.0007), with the antimicrobial paint exhibiting a 59% lower median bioburden than the control paint.

### Hospital locker room study

Surfaces evaluated as part of the pre-intervention organic contamination screen at the hospital included walls, bed rails, call buttons, chair armrests, tray tables, drawer handles, and lockers used by hospital staff. The walls had an organic contamination level (average = 81 RLU, n = 94) similar to that found on higher-touch surfaces such as bed rails, call buttons, tray tables, drawer handles, IV poles, and arm rests (average = 87 RLU, n = 264). Of the non-wall surfaces screened, lockers had higher levels of organic material (average = 100 RLU, n = 189) than other surfaces (average = 87 RLU, n = 409). This suggested that the locker rooms had higher levels of either organic, non-microbial or microbial contamination, presumably resulting from the lack of disinfection practices relative to surfaces in admitted patient areas. Measurable levels of bioburden on the locker room surfaces were confirmed, with higher levels on the lower locker vents (average = 2,437 CFU, n = 10) located closer to the ground (~1 foot above floor) than on upper locker surfaces (average = 399 CFU, n = 10) (above locker handle, ~3 feet above floor). This finding is consistent with prior measurements of bacterial contamination versus wall height (**Fig 1**).

Due to the skewness in the data and the expectation of unequal variance between paint types, median bioburden counts were analyzed (**Fig 3**). The median bacterial count for the men's locker surfaces with the antimicrobial paint was 595 CFU/100 cm$^2$; the median count with the control paint was 1,611 CFU/100 cm$^2$. This difference in median bioburden was

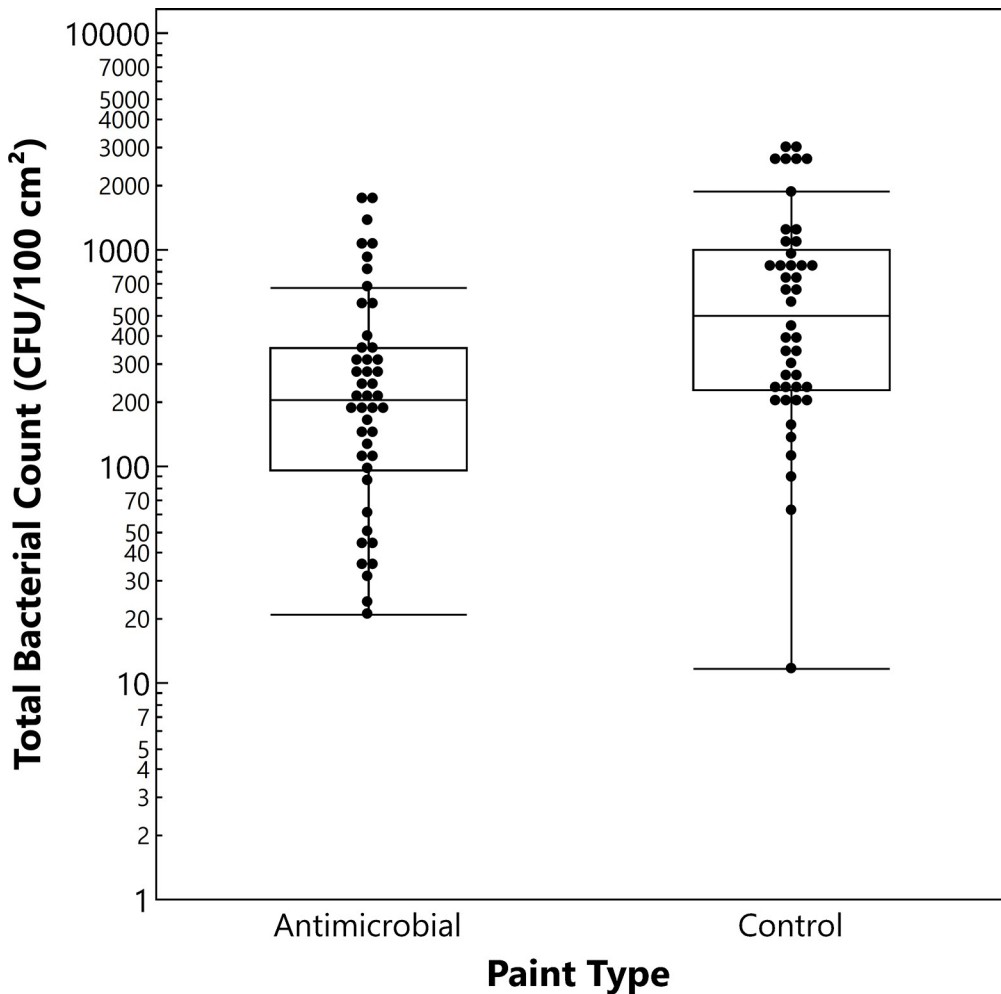

**Fig 2. HPC bacterial counts on preschool restroom walls with copper-containing antimicrobial painted and control painted surfaces.** The box and whisker plot displays the count distribution: Median (line within box), quartiles (box top and bottom lines = upper and lower limits of 1st and 4th quartiles, respectively), and outliers (beyond whiskers).

statistically significant ($P$ = 0.0002), with the antimicrobial paint exhibiting a 63% lower median bioburden than the control paint. The median bioburden on the locker surfaces in the women's locker room was higher than in the men's locker room for both the antimicrobial paint (2,401 CFU/100 cm$^2$) and control paint (2,031 CFU/100 cm$^2$) and was not significantly different between the paint types ($P$ = 0.057). The difference in median bioburden between locker rooms was statistically significant for the antimicrobial painted surfaces ($P<0.0001$), but there was no difference for the control painted surfaces ($P$ = 0.33). The antimicrobial painted surfaces exhibited a similar trend in the men's locker room for the spiked *E. coli* control levels (average log recovered 3.29±0.26 CFU/100 cm$^2$) compared to the control painted surfaces (average log recovered 4.55±0.93 CFU/100 cm$^2$); no differences were observed for the women's locker room. The reasons for the lack of differentiation between the two painted surfaces in the women's locker room is unclear but potential reasons are discussed later. Median microbial measurements, further broken down by month for the men's locker surfaces, are summarized in **Fig 4**. The data show relatively consistent reduction in microbial burden across the study period.

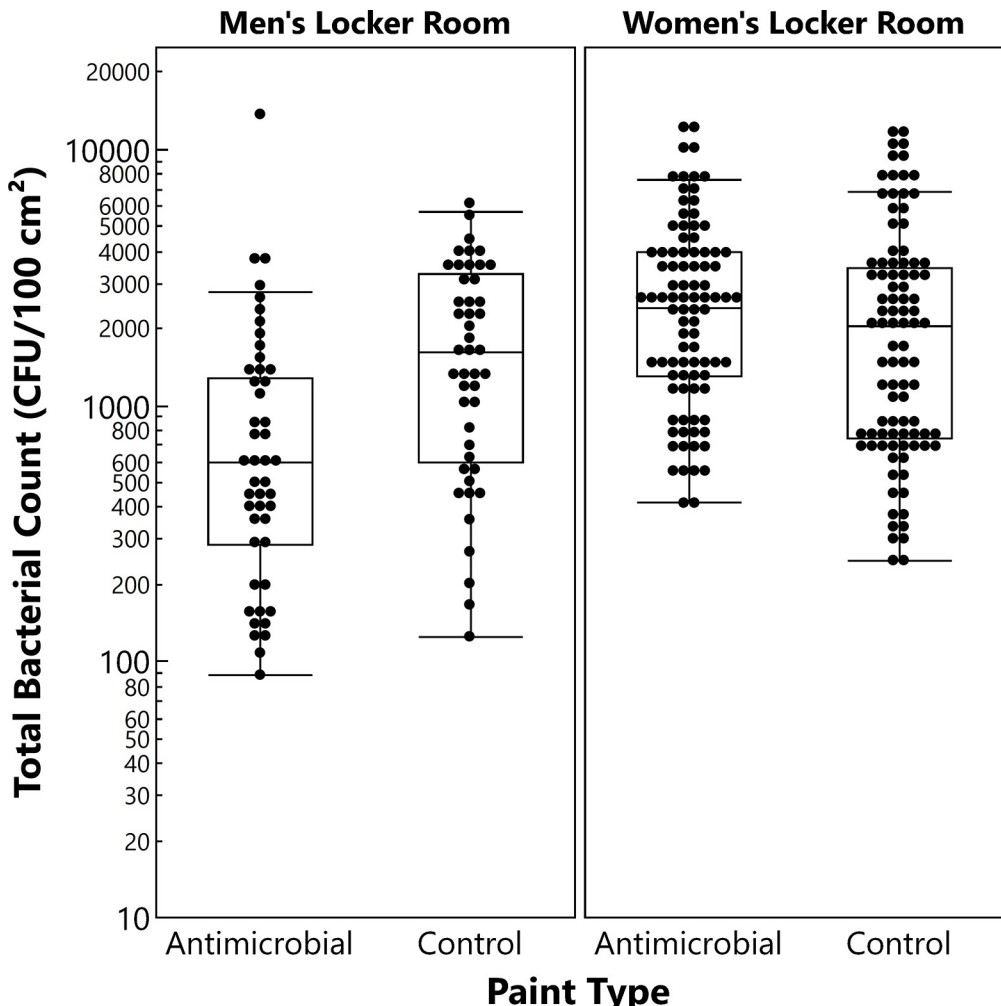

**Fig 3. HPC bacterial counts on hospital lockers with copper-containing antimicrobial-painted and control-painted surfaces.** The box and whisker plot displays the count distribution: Median (line within box), quartiles (box top and bottom lines = upper and lower limits of 1st and 4th quartiles, respectively), and outliers (beyond whiskers).

In the men's locker room, two antimicrobial painted lockers and six control painted lockers were positive for coliforms. In the women's locker room, six antimicrobial painted lockers and 14 control painted lockers were positive for coliforms (**Table 1**). A reduction in coliform counts was observed on antimicrobial paints compared to control paints, but the sample size was too small to determine significance. Only three samples were positive for *E. coli* during the entire study. One was from an unpainted locker handle (control) in the women's locker room (with 1.78 MPN/100 cm$^2$) during month six. The other two positive samples were collected from the men's locker room in month five–one from a locker with antimicrobial paint and one from a locker with control paint. Both were at the detection limit for the assay (0.23 MPN *E. coli*/100 cm$^2$).

## Summary statistics with upper quartile bacterial counts

Summary statistics from studies conducted at the preschool restroom and hospital locker rooms are shown in **Table 1**. The upper quartile bacterial counts (highest 25%) were analyzed

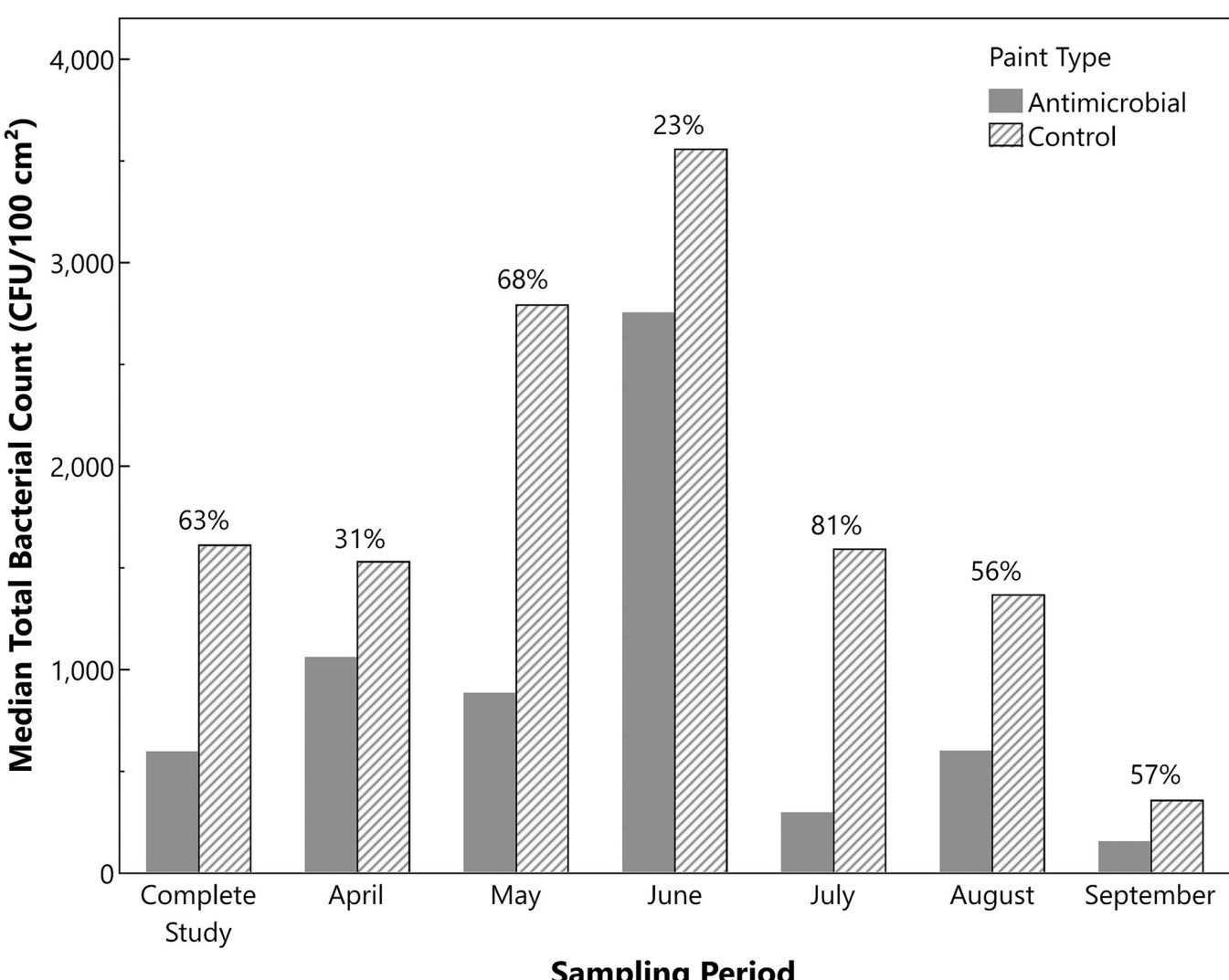

**Fig 4. Median bacterial counts over time from the men's hospital locker rooms recovered from the copper-containing antimicrobial painted surfaces (solid) and the control painted surfaces (striped).** The corresponding percent reductions in median bacterial counts due to the antimicrobial surfaces are listed for each month.

separately since surfaces with higher contamination levels are the main contributors to both the risk of cross-contamination and exposure to pathogens. The upper quartile HPC counts followed the same trend as the total counts, with the antimicrobial paint showing a 63% reduction and 47% reduction in bacterial counts compared to the control paint in the preschool restroom and hospital men's locker room, respectively. There was not a statistically significant difference in the upper quartile bacterial counts between the two painted surfaces for the women's locker room.

## Discussion

In response to the COVID-19 pandemic, the EPA created a new category of residual supplemental antimicrobial products that were allowed to make both bactericidal and virucidal claims [12]. For registration, products are tested for efficacy against core organisms across the germ hierarchy (e.g., enveloped/non-enveloped viruses, Gram-positive/Gram-negative

**Table 1. Summary statistics from interventional studies.**

| Intervention Location | Preschool Restroom Walls | | Hospital Locker Room | | | |
|---|---|---|---|---|---|---|
| | | | Men's | | Women's | |
| Paint Type | Antimicrobial | Control | Antimicrobial | Control | Antimicrobial | Control |
| Total Samples | 45 | 44 | 47 | 46 | 46 | 47 |
| Median Bacterial Count (CFU/100 cm$^2$) | 204 | 494 | 595 | 1611 | 2401 | 2031 |
| P | 0.0007 | | 0.0002 | | 0.057 | |
| Percent Reduction | 59% | | 63% | | 0% | |
| Upper Quartile Bacterial Count (CFU/100 cm$^2$) | 915 | 2499 | 2067 | 3897 | 6269 | 6672 |
| P | 0.003 | | 0.006 | | 0.88 | |
| Percent Reduction | 63% | | 47% | | 6% | |
| Positive Coliform Result[a] | N/A | | 2 of 47 | 6 of 46 | 6 of 46 | 14 of 47 |
| Percent Positivity Rate | N/A | | 4.3% | 13.0% | 13.0% | 25.5% |

[a]Coliform testing was not conducted in the preschool restroom.

bacteria). Claims of efficacy against human pathogens can only be obtained for products that pass the EPA copper protocol (introduced in 2008 [13]) because it is a more realistic simulation of contamination than traditional antimicrobial tests (e.g., JIS Z 2801, ISO 22196). Furthermore, products must undergo wear testing to demonstrate antimicrobial efficacy after wiping with cleaners simulating five years of cleaning. The EPA-registered antimicrobial paint used in these evaluations has claims for killing 99.9% of the following bacteria/viruses within two hours of exposure: *Staphylococcus aureus*, *Pseudomonas aeruginosa*, *Enterococcus faecium*, *Klebsiella pneumoniae*, *Acinetobacter baumannii*, *Enterobacter aerogenes*, methicillin-resistant *Staphylococcus aureus* (MRSA), vancomycin-resistant *Enterococcus* (VRE), *Escherichia coli*, *Salmonella*, feline calicivirus, and SARS-CoV-2 virus. This paint is also listed on the EPA List N Appendix: Supplemental Residual Antimicrobial Products for Coronavirus (COVID-19) [9].

Schmidt et al. [6] studied the effectiveness of intervention with metallic copper alloy surfaces in healthcare settings, reporting an 83% reduction in bioburden for objects formerly covered with plastic, wood, or stainless-steel. Importantly, studies in hospital intensive care unit rooms employing copper alloy surfaces showed a 58% decrease in hospital-acquired infections compared to standard rooms [6,7]. In the present study with the first reported field trial of a copper-containing supplemental residual antimicrobial paint, an average microbial bioburden reduction of 59% on the preschool restroom walls and 63% in the men's locker room was observed when compared to control paints. The measured bioburden reduction was significant, though lower than reported by Schmidt and co-workers for metallic copper [6]. The results also demonstrated that the antimicrobial paint reduced the upper quartile of microbial contamination, which is important for mitigating risk. Such peak pathogen levels drive the overall risk since they increase the odds of infection [14]. The upper quartile of bacterial counts was reduced by 63% in the preschool restroom and by 47% in the men's locker room. Further, fewer positive coliform counts were observed in both the men's and women's locker rooms when using the antimicrobial paint.

Studies were conducted over 4–6 months in both locations. There was an insufficient number of timepoints for a long-term statistical analysis of product performance; however, the

median bioburden counts for the men's locker room showed a consistent trend. The antimicrobial paint had a lower mean bioburden than the control paint at each sampling timepoint and there were observed variations in the total bioburden over time. The reduction at month four could be explained by the fact that the surfaces were disinfected with isopropyl alcohol wipes one week prior to sample collection, removing debris and possible contamination. Irrespective of absolute bioburden, the results also showed that the reduction in microbial contamination due to the antimicrobial paint was maintained over the study period, suggesting a long-lasting nature of its antimicrobial potency.

While two of three intervention areas demonstrated a statistically significant decrease in bioburden on the antimicrobial painted surface, no reduction was observed for the women's locker room for the median bioburden or upper quartile of bioburden. The reason for this difference relative to the men's locker room is unclear. A factor that could have influenced the results is the level of soil found on painted surfaces. Interfering substances like dirt, dust, and organic contaminants from cosmetics or personal care products could prevent organisms from contacting the painted surface and limit its efficacy. Differences in the level of bioburden present in environments occupied by men and women have been observed in other studies, such as in offices [15] and university dormitories [16].

A strength of this study was the ability to compare the supplemental residual antimicrobial paint product at the same time and in the same environment as the control paint. This eliminated study variables associated with an interventional study focused on a pre- and post-intervention analysis. The evaluation of multiple locations over 4–6 months further supports the reliability of the findings. In addition, a large surface area was sampled (~232–619 cm$^2$) and a culture medium designed to optimize the recovery of a large range of bacterial types and environmentally-stressed bacteria was used.

Although microbial contamination of low-touch areas is considered a lower risk for pathogen transmission, they can still serve as a reservoir of disease-causing microorganisms. This is especially true of areas outside of patient rooms (e.g., break rooms, waiting rooms) that do not receive the same degree of cleaning. Pathogens on low-touch areas may also be resuspended/re-aerosolized by human activity, resulting in contamination of more frequently-touched surfaces [17].

## Conclusions

The results from this study demonstrate that there are detectible levels of contamination found on large-area surfaces like walls and lockers and that contamination levels can be reduced through use of a supplemental residual antimicrobial coating. The application of antimicrobial paints to walls and other large-area surfaces could be a practical and cost-effective addition to comprehensive cleaning/disinfection practices. A distinct advantage of using paint is its broad applicability to a variety of surface types and its large color palette, enabling everyday objects and surfaces to be imparted with antimicrobial efficacy with minimal compromise to aesthetics and décor. As this was the first field study of a supplemental residual antimicrobial paint, additional studies are recommended to further assess the potential benefits of using antimicrobial paints as part of a comprehensive disinfection plan. Antimicrobial paints are ideally suited for addressing neglected surfaces and may lead to meaningful reductions in contamination.

## Supporting information

**S1 Dataset. S1 Raw dataset.** Raw dataset for all figures in the manuscript.
(XLSX)

## Author Contributions

**Conceptualization:** Jennifer Hiras, Kelly R. Bright, Anthony G. Frutos, Charles P. Gerba, Joydeep Lahiri.

**Formal analysis:** Jennifer Hiras, Kelly R. Bright, Alexander E. McInroy.

**Investigation:** Jennifer Hiras, Kelly R. Bright, Jackie L. Kurzejewski, Jon Q. Lehman.

**Methodology:** Jennifer Hiras, Kelly R. Bright, Alexander E. McInroy.

**Project administration:** Jennifer Hiras, Kelly R. Bright, Anthony G. Frutos.

**Writing – original draft:** Jennifer Hiras, Kelly R. Bright, Mark R. Langille, Joydeep Lahiri.

**Writing – review & editing:** Jennifer Hiras, Kelly R. Bright, Alexander E. McInroy, Anthony G. Frutos, Mark R. Langille, Charles P. Gerba, Joydeep Lahiri.

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
