## [Decision Letter · Decision Letter 0]

16 Jun 2024

PONE-D-24-06372Reduction of bioburden on large area surfaces through use of a supplemental residual antimicrobial paintPLOS ONE

Dear Dr. Lahiri,

Thank you for submitting your manuscript to PLOS ONE. After careful consideration, we feel that it has merit but does not fully meet PLOS ONE’s publication criteria as it currently stands. Therefore, we invite you to submit a revised version of the manuscript that addresses the points raised during the review process.

We look forward to receiving your revised manuscript.

Kind regards,

Amitava Mukherjee, ME, Ph.D.

Academic Editor

PLOS ONE

Journal Requirements:

"Conflicts of interest: Jennifer Hiras, Jackie L. Kurzejewski, Alexander E. McInroy, Anthony G. Frutos, Mark R. Langille, and Joydeep Lahiri are employees at Corning Incorporated; Corning has commercialized the copper-glass ceramic additive (Corning® Guardiant®) used in the antimicrobial paint reported. This study was funded by Corning Incorporated and the University of Arizona received fees for services rendered for the processing and assay of the collected samples for the detection and quantification of HPC bacteria, coliforms, and Escherichia coli."

We note that one or more of the authors are employed by a commercial company: Corning Incorporated

A. Please provide an amended Funding Statement declaring this commercial affiliation, as well as a statement regarding the Role of Funders in your study. If the funding organization did not play a role in the study design, data collection and analysis, decision to publish, or preparation of the manuscript and only provided financial support in the form of authors' salaries and/or research materials, please review your statements relating to the author contributions, and ensure you have specifically and accurately indicated the role(s) that these authors had in your study. You can update author roles in the Author Contributions section of the online submission form.

B. Please also provide an updated Competing Interests Statement declaring this commercial affiliation along with any other relevant declarations relating to employment, consultancy, patents, products in development, or marketed products, etc.  

Within your Competing Interests Statement, please confirm that this commercial affiliation does not alter your adherence to all PLOS ONE policies on sharing data and materials by including the following statement: ""This does not alter our adherence to  PLOS ONE policies on sharing data and materials.” (as detailed online in our guide for authors http://journals.plos.org/plosone/s/competing-interests) . If this adherence statement is not accurate and  there are restrictions on sharing of data and/or materials, please state these. Please note that we cannot proceed with consideration of your article until this information has been declared.

Reviewers' comments:

Reviewer's Responses to Questions

**Comments to the Author**

1. Is the manuscript technically sound, and do the data support the conclusions?

Reviewer #1: Yes

Reviewer #2: Yes

2. Has the statistical analysis been performed appropriately and rigorously? 

Reviewer #1: Yes

Reviewer #2: Yes

3. Have the authors made all data underlying the findings in their manuscript fully available?

Reviewer #1: Yes

Reviewer #2: Yes

4. Is the manuscript presented in an intelligible fashion and written in standard English?

Reviewer #1: Yes

Reviewer #2: Yes

5. Review Comments to the Author

Reviewer #1: Dear authors,

Thank you for this interesting manuscript. I have just some comments:

Line 71: Please delete "CFU" (mentioned in the next line again)

Line 103/104: Why is the control paint used for 24 lockers in women's locker room and the antimicrobial paint for 41 lockers?

Line 163: Please defined "RLU" (first occurrence)

Reviewer #2: The article provides a comprehensive description of the findings of the study. The methodology employed is thoroughly explained. The results are presented in a clear and concise manner, and are supported by relevant data. The discussion section of the article presents a well-informed analysis of the results, drawing on relevant literature and theoretical frameworks.

6. PLOS authors have the option to publish the peer review history of their article (what does this mean?). If published, this will include your full peer review and any attached files.

Reviewer #1: No

Reviewer #2: No

---

## [Author Response · Author response to Decision Letter 0]

17 Jul 2024

We have responded to the editor and reviewer comments in the cover letter, as instructed.

---

## [Editor Report · Decision Letter 1]

22 Jul 2024

Reduction of bioburden on large area surfaces through use of a supplemental residual antimicrobial paint

PONE-D-24-06372R1

Dear Dr. Lahiri,

We’re pleased to inform you that your manuscript has been judged scientifically suitable for publication and will be formally accepted for publication once it meets all outstanding technical requirements.

Kind regards,

Amitava Mukherjee, ME, Ph.D.

Academic Editor

PLOS ONE
---

## [Editor Report · Acceptance letter]

29 Aug 2024

PONE-D-24-06372R1 

PLOS ONE

Dear Dr. Lahiri, 

I'm pleased to inform you that your manuscript has been deemed suitable for publication in PLOS ONE. Congratulations! Your manuscript is now being handed over to our production team.

Kind regards, 

on behalf of

Professor Dr. Amitava Mukherjee 

Academic Editor

PLOS ONE